# Ref-MEF: Reference-Guided Flexible Gated Image Reconstruction Network for Multi-Exposure Image Fusion

**DOI:** 10.3390/e26020139

**Published:** 2024-02-03

**Authors:** Yuhui Huang, Shangbo Zhou, Yufen Xu, Yijia Chen, Kai Cao

**Affiliations:** College of Computer Science, Chongqing University, Chongqing 400044, China

**Keywords:** multi-exposure fusion, image processing, computational photography, convolutional neural networks

## Abstract

Multi-exposure image fusion (MEF) is a computational approach that amalgamates multiple images, each captured at varying exposure levels, into a singular, high-quality image that faithfully encapsulates the visual information from all the contributing images. Deep learning-based MEF methodologies often confront obstacles due to the inherent inflexibilities of neural network structures, presenting difficulties in dynamically handling an unpredictable amount of exposure inputs. In response to this challenge, we introduce Ref-MEF, a method for color image multi-exposure fusion guided by a reference image designed to deal with an uncertain amount of inputs. We establish a reference-guided exposure correction (REC) module based on channel attention and spatial attention, which can correct input features and enhance pre-extraction features. The exposure-guided feature fusion (EGFF) module combines original image information and uses Gaussian filter weights for feature fusion while keeping the feature dimensions constant. The image reconstruction is completed through a gated context aggregation network (GCAN) and global residual learning GRL. Our refined loss function incorporates gradient fidelity, producing high dynamic range images that are rich in detail and demonstrate superior visual quality. In evaluation metrics focused on image features, our method exhibits significant superiority and leads in holistic assessments as well. It is worth emphasizing that as the number of input images increases, our algorithm exhibits notable computational efficiency.

## 1. Introduction

Leveraging high dynamic range (HDR) imaging technology allows for a more intricate and expanded color palette in different media forms, including videos and images. However, the dynamic range captured by traditional image sensors like CCD or CMOS deviates significantly from the actual scene due to their ability to capture only low dynamic range images. Direct acquisition of HDR images imposes the need for more advanced and costly equipment. As a cost-effective alternative, multi-exposure image fusion (MEF) offers substantial advantages. It creates a comprehensive exposure map of a scene by merging low dynamic range (LDR) images obtained under various exposure settings, thereby producing an HDR image and enhancing visual perception. Moreover, the fused image retrieves and accentuates details otherwise lost due to underexposure or overexposure.

The extensive study of MEF is attributable to its effectiveness in generating high-quality images. Traditional MEF techniques [1,2,3] necessitate the manual creation of feature extraction methods and fusion rules. However, this process is not only laborious and intricate but also demonstrates limited robustness under varying input conditions. Recently, Convolutional Neural Networks (CNNs) have made significant progress in low-level tasks [4,5,6], such as image reconstruction and fusion, due to their robust representational capabilities. DeepFuse [7], a pioneering application of CNNs in the MEF domain, leverages the Multi-Exposure Structural Similarity Index (MEF-SSIM) [8] as a loss function, thereby overcoming the constraints of manual feature design.

Existing MEF algorithms predominantly employ feature concatenation [9,10], which restricts the input exposure sequence length to the number of input neurons, limiting fusion to merely two or three images. This insufficiency in exposure sequence numbers hinders the achievement of comprehensive exposure coverage, highlighting the necessity for MEF algorithms adept at handling sequences of infinite length. To circumvent this issue, most methods adopt a model nesting approach, which inadvertently leads to a linear escalation in model fusion time, undermining real-time performance. Although DeepFuse’s [7] feature addition method overcomes this numerical limit, it may result in significant value range discrepancies when dealing with image sequences of varying lengths. The mean or average method for feature fusion proposed by IFCNN [11] indeed mitigates this issue, but it can inadvertently lead to significant feature degradation.

MEF-Net [12] proposes a single-input network that eliminates the connection between images in the same exposure sequence. It calculates weights for each image and uses a weighted fusion method. However, this approach assumes that the image sequences are independent of each other, ignoring the redundancy and complementarity of images captured under the same exposure conditions. Consequently, this assumption results in a loss of detail and color information.

This paper introduces an MEF method capable of accommodating an uncertain quantity of inputs, facilitating the recovery of static scenes while preserving color information. The contributions of this paper are outlined as follows:We introduce a reference-guided exposure correction (REC) module, rooted in the reference image, which incorporates both channel and spatial attention. This module significantly alleviates color discrepancies and artifacts in the fused image.Within the reconstruction network, we deploy an exposure-guided feature fusion (EGFF) module designed to standardize features of varying lengths to a common dimension. Simultaneously, we enhance the gated context aggregation network (GCAN) to efficiently gather deeper contextual information, preserve spatial resolution, and effectively suppress grid artifacts.An optimized loss function is proposed to uphold lighting and texture information, ensuring the comprehensive retention of color information from the exposure sequence images.

Ultimately, we introduce Ref-MEF, a novel and flexible MEF algorithm, specifically designed for multi-exposure image fusion with unpredictable input quantities and resolutions. We carry out end-to-end training on static multi-exposure image sequences processed by registration algorithms [13,14,15] using the loss function we develop. Several experiments indicate that, compared to most MEF algorithms, our method displays a significant advantage, both in subjective visual effects and objective evaluation metrics. Additionally, our method holds a clear computational time advantage when dealing with an increased number of exposure sequences.

## 2. Related Work

In this section, we begin with an explanation of different existing MEF methodologies, followed by an incisive review of relevant research endeavors closely related to our own work.

### 2.1. Existing MEF Methods

MEF algorithms are categorized into dynamic and static types based on the inherent characteristics of the shooting scene. Designing dynamic MEFs requires addressing object motion and camera shake to minimize ghosting effects. Predominantly, these algorithms involve motion detection, image registration, and image reconstruction. Image registration is crucial in counteracting camera shake. Starting with image alignment and registration enables the conversion of dynamic MEFs into static ones. This paper focuses primarily on static MEFs.

Static MEF algorithms are categorized into traditional and deep learning-based approaches. The traditional methods are subdivided into spatial domain-based and transformation domain-based methods. Within the spatial domain, the strategies include: (1) Pixel-level techniques adhere to a paradigm of fusion weighting, at the heart of which lies the development of efficacious weighting functions. For example, Liu et al. [16] proposed an MEF approach that employed dense Scale-Invariant Feature Transform (SIFT) descriptors and guided filtering. This method constructed a weight map informed by spatial consistency, exposure quality, and local contrast, leveraging guided filters to refine weights and mitigate noise. The dense SIFT descriptors gauged activity levels, rendering the method apt for dynamic scenes. (2) Patch-based techniques segment images into patches to compose the fused imagery. A notable method by Ma et al. [2] segmented patches into distinct components of signal strength, signal structure, and mean intensity. Through assessing patch intensity and exposure, these components were integrated to form the fused patches, which were subsequently amalgamated into the final fused image. SPD-MEF [17], developed by Ma’s team, employed signal structure decomposition of patches to compute consistency maps for ghosting elimination. APS-MEF [18] improves dynamic contrast by integrating image patch structure decomposition, image cartoon texture decomposition, and the structural similarity index. (3) Optimization-based techniques, exemplified by MEF-Opt [8], aimed to optimize the MEF-SSIMc [8] index within the image domain. Although there is no universally accepted benchmark for appraising MEF algorithms, this method’s forte is its adaptability to metrics that more accurately assess fusion outcomes. Notwithstanding, it necessitated extensive computational time due to its iterative global optimization in the image domain.

Regarding transformation domain-based methods, they typically encompass domain transformation, fusion within the transformed domain, and the subsequent inverse transformation. Effective fusion strategies are enacted within the transformed domain, with prevalent techniques involving multi-scale transformations, such as principal component analysis, wavelets, pyramids, and dense invariant feature transformations. Karakaya et al. [1] introduced PAS-MEF, grounded in principal component analysis, adaptive exposure, and saliency maps, harnessing pyramid decomposition for fusion. This preserved the richest information from each exposure image and boasted a rapid processing speed.

Recent advancements in the application of Convolutional Neural Networks (CNNs) in the MEF field underscored their remarkable image representation capabilities. These capabilities eliminated the need for complex manual fusion strategy design found in traditional methods, thus exhibiting high robustness. Deep learning methods predominantly fell into two categories: unsupervised and supervised. DeepFuse [7], the pioneer in incorporating deep learning technology in the MEF field, used an unsupervised measure, MEF-SSIM [19], as the loss function. This breakthrough approach transformed RGB images into YCbCr color space, focusing fusion design solely on the Y channel, whereas the network structure remained simple; the concept of unsupervised metric fusion and the design on the Y channel sparked various subsequent studies. FusionDN [20] and U2Fusion [21] consolidated multiple image tasks into one densely connected network, employing elastic weight merging. This approach enabled a single network model to tackle multiple image fusion tasks, inclusive of multi-modal, multi-exposure, and multi-focus. U2Fusion, building on FusionDN, refined the information preservation distribution strategy and loss function. FusionDN’s losses comprised structural similarity Index (SSIM), perceptual loss, and gradient loss. Conversely, U2Fusion replaced the gradient loss with mean squared error (MSE) loss, effectively reducing the brightness bias in fusion outcomes. ACE-MEF [22] preserves the texture and environmental light details in the source image by constructing the clarity preservation network (CPN) network. The illumination adjustment network (IAN) network corrects locally severe exposure, and a specific MEF loss is built to guide CPN in adaptively retaining light and detail information in clear areas.

Supervised methods require a significant amount of ground truth for training; however, a large-scale ground truth dataset in the MEF field is currently non-existent. Consequently, it becomes necessary to either manually generate or simulate “ground truth”. Wang et al. [23] generated “ground truth” by manipulating the pixel intensities of the ILSVRC 2012 validation image set [24]. Nevertheless, the authenticity of the ground truth generated by this method could be disputable as it may not accurately reflect the actual shooting effect of real-world scenes. An alternate approach involves the use of outcomes from prior algorithms to simulate “ground truth”. For instance, the SCIE dataset [25] not only offered 500 sets of multi-exposure image sequences from static scenes but also presented fusion results from 13 representative algorithms. IFCNN [11] employed a dual-branch architecture with shared weights, merged features extracted from convolution based on element mean, and proceeded to train the model utilizing perceptual loss and the fundamental loss between the source image sequence and the simulated “ground truth”. CFMSAN [26] employs a multi-scale attention-guided network to extract features across various scales, generating attention weight maps of multiple sizes. These weight maps guide the fusion result generation. The model undergoes training on the SCIE [25] dataset through a supervised manner.

Apart from CNNS, an alternative category of methodologies based on Generative Adversarial Networks (GANs) is employed to tackle the MEF issue. Taking inspiration from the FusionGAN [27] model, which effectively fuses infrared and visible images, GANFuse [28] enhances adversarial learning by increasing the number of discriminators. This augmentation facilitates the assimilation of valuable information from pairs of images with extreme exposure. MEF-GAN [9] and Chen [29] et al. establish a GAN wherein the generator produces fused images, and the discriminator determines the distinguishability of the fusion results from the forged “real labels”. To achieve feature fusion based on attention and long-term dependencies, MEF-GAN [9] incorporates a self-attention mechanism within the generator. However, the imposed constraints on MEF-GAN (requiring image dimensions to be multiples of 8,somewhat curtail its versatility, rendering it unsuitable for arbitrary resolutions. It is noteworthy that Chen [29] et al. integrate a Homography network into their model to compensate for camera motion. This incorporation renders their method applicable to dynamic MEF fusion scenes afflicted by camera jitter.

TransMEF [30] presently introduced a self-supervised, multi-task learning framework premised on the Transform architecture. This framework underwent training on extensive natural datasets, circumventing the need for ground truth. However, the MS-COCO dataset [31], primarily utilized for object detection and segmentation, fell short of meeting rigorous requirements due to its inherent image dynamic range and texture detail information. The brightness and texture data gathered from the MS-COCO dataset failed to adhere to the high-quality standards set forth for MEF image reconstruction. Moreover, the method TransMEF employed for the reconstruction of brightness channels was incapable of restoring the color information in the scene with precision.

### 2.2. Most Relevant Work

Our research work is intimately related to MEF-Net [12]. The MEF-Net also has the ability to solve network flexibility issues and can receive inputs of varying lengths. This network achieved this by constructing a single-input network, setting the number of input images equal to the batch size, disconnecting the correlation between image sequences, independently calculating the weights of each input, and performing image weighting operations. MEF-Net only used the brightness channel of the image for training, and only took MEF-SSIM [19] as the loss function, so it cannot retain the chroma information in the captured scene.

In contrast, our network fully utilizes the relationship between source image sequences and builds a weight map that has been smoothed by Gaussian after acquiring exposure information between source sequences, forming regional characteristics. By weighting and merging the corrected features, we send a fixed number of feature channels into the reconstruction network for training. Ref-MEF combines MEF-SSIMc [8] and gradient fidelity items to construct the loss function, not only preserving the chroma information in the scene as much as possible but also retaining the high-frequency components and texture integrity of the fusion results.

It is worth noting that both approaches adopt the context aggregation network (CAN) [32]. The CAN in MEF-Net was used to generate a weight map. Our enhancements to the CAN include the introduction of separable convolutions before dilated convolutions, thereby substantially strengthening the interdependence of the input units in the dilated convolutions and significantly diminishing grid artifacts. Furthermore, we have integrated a gating structure, combined multi-scale information, and employed global residual learning (GRL) for image reconstruction learning.

## 3. Methodology

We present Ref-MEF, a highly flexible and superior multi-exposure image fusion method. As illustrated in Figure 1, our approach can accommodate multi-exposure image sequences of varying lengths by first converting them into feature maps through the encoder section. We then refine these features using the reference-guided exposure correction (REC) module. In the reconstruction network section, we utilize the exposure-guided feature fusion (EGFF) module to ensure adaptability to different exposure levels. Additionally, we employ the reference-guided exposure correction (GCAN) and global residual (GRL) connections to facilitate efficient network reconstruction and information enhancement, ultimately improving the quality of image fusion. Finally, the reconstructed feature maps are decoded back into the original image space, producing a high-quality fused image.

### 3.1. Reference-Guided Exposure Correction

As depicted in Figure 1, initial processing is performed on the input sequence images of multiple exposures. This procedure is facilitated by a universal encoder, which consists of three convolution layers and is tasked to map the images from the LDR domain to a distinct feature space for extraction and pre-processing of image features. For uniform feature space distribution and semantic information across LDR images, we have utilized a parameter-sharing approach to facilitate comparisons. This technique uses the identical encoder for all LDR images, providing superior operational efficiency and substantial reduction in network parameters compared to multiple autonomous encoders. Despite the shared-weight encoder’s ability to efficiently extract generic image features, it falls short in characterizing the relationships among images with varying exposure levels and their associated features. Therefore, to improve image feature representation precision and foster a more profound study of exposure sequence interrelations, we introduce the reference-guided exposure correction (REC) module.

To determine the reference image essential for the REC module, we utilized the Peak-to-Average Ratio (PAR) for histogram quantization. The PAR signifies the ratio between the peak and valley of the histogram, serving as an indicator of the histogram’s uniformity, specifically the uniform distribution of image brightness. An image with a diminished PAR generally suggests the absence of conspicuous overexposed or underexposed regions, as overexposure and underexposure result in abnormally elevated or diminished peaks and valleys in the histogram. Opting for such an image as a reference mitigates the introduction of unwarranted distortions during the fusion process. Furthermore, a lower PAR is frequently linked to superior preservation of image details, contributing to the overall retention of scene intricacies. We assume that the brightness has been sorted from low to high, and the reference image’s index is defined as: (1)$$r=\arg\min PAR_{i},\mathrm{for}\;i=1,2,\ldots ,K,$$
where argmin(·) is a search operator used to locate the index of the image with the smallest PAR value within the exposure sequence, where *K* represents the current length of the exposure sequence. Here, *r* divides the image sequence into two parts: the underexposed images $L_{u}$ and the overexposed images $L_{o}$, with the corresponding encoders extracting the features as $E_{u}$ and $E_{o}$, respectively.

As depicted in Figure 2, the REC module combines spatial attention and channel attention to adjust the impact of non-reference features on image reconstruction by utilizing spatial information, exposure intensity information, and channel color information of the reference image.

Consider the underexposed feature $E_{j}$, an element of $E_{u}$. The $REC_{u}\left(\cdot \right)$ refines $E_{j}$’s contribution to image reconstruction by utilizing spatial, exposure, and color information from the reference image. The module automatically adjusts the weight $A_{j}$ of the LDR image, specifically for those images with relatively lower exposure levels. This weight encapsulates not only the image’s own exposure information but also integrates related content from the reference image. This integration allows the enhancement of non-reference images through the reference image’s information, thus uncovering the relationships among images with differing exposure degrees.

In a similar way, we can also obtain the feature $F_{o}$ adjusted by $REC_{o}\left(\cdot \right)$ in the overexposed area. To ensure that the network can cope with the image reconstruction problem of the exposure sequence of indefinite length, we adopt a parameter-sharing strategy for its implementation module.
(2)$$\begin{array}{c} F_{j}=E_{j}⨂A_{j}=E_{j}⨂REC_{i}\left(concat\left(E_{j},E_{r}\right)\right)\\ =e\left(L_{j}\right)⨂REC_{i}\left(\left[e\left(L_{j}\right);e\left(L_{r}\right)\right]\right). \end{array}$$

$REC_{i}\left(\cdot \right)$, comprising $REC_{u}\left(\cdot \right)$ and $REC_{o}\left(\cdot \right)$, employs “concat” for channel dimension concatenation, with e(·) denoting the encoder and ⨂ indicating element-wise multiplication.

### 3.2. Reconstruction Network

#### 3.2.1. Exposure-Guided Feature Fusion

To integrate the variably quantified correction features derived from the REC module into the reconstruction network for further processing, it is crucial to amalgamate and stabilize the features to a designated number of channels. Existing feature-merging methods, employing average-featured or maximum value selection [11], lead to suboptimal network flexibility and feature degeneration. Thus, we propose an exposure-guided feature fusion (EGFF) strategy.

This innovative method is built upon the principle that during the merging process, the weights corresponding to well-exposed regions should be accentuated, whereas those associated with poorly exposed regions should be significantly reduced. After establishing this objective, we use average brightness as a metric, setting the ideal brightness level away from 0 (indicating underexposure) and 1 (indicating overexposure), specifically at 0.5.

Following a Gaussian curvature measurement, pixel-level weights are derived. Nonetheless, given that $F_{j}$ is a regionally distinctive feature with asymmetric spatial information, a direct weighting process is not feasible. Instead, we necessitate a subsequent transformation process to endow the weights with region-centric properties. Concurrently, the unity of features should maintain receptive fields similar to or the same as $F_{j}$. Using Gaussian curvature as a measuring scale, $w_{{\mathrm{gaussian}}_{j}}$ are obtained as outlined in Equation (Equation 3).
(3)$$\begin{array}{c} w_{{\mathrm{gaussian}}_{j}}=\exp\left(-\frac{∥{\bar{L}}_{j}{-0.5∥}_{F}^{2}}{2\sigma^{2}}\right),\;\sigma =0.2,\;1\leq j\leq K, \end{array}$$
wherein ${\bar{L}}_{j}$ denotes the mean brightness of the image, computed as the numeric mean of the three color channels of the image. The term ${∥\cdot ∥}_{F}$ signifies the computation of the Frobenius norm, whereas *K* designates the length of the present exposure sequence.

We implement a Gaussian filtering procedure on the weight map, utilizing a Gaussian filtering kernel with standard deviation $\sigma^{*}$ and Gaussian convolution kernel $G_{\sigma^{*}}$. This procedure enhances the weight’s regional attributes, redresses its uneven distribution and smoothness, and enhances its resistance to noise interference. The entire process can be demonstrated as a convolution operation, with the Gaussian kernel acting as a filter to enhance the quality of the weight data.
(4)$$\begin{array}{c} w_{i}=G_{\sigma^{*}}*w_{{\mathrm{gaussian}}_{j}},\;1\leq j\leq K. \end{array}$$

Finally, we employ a normalization process to the weights, accompanied by a pixel-level weighting methodology, to accrue a feature map $F^{\prime }$ post the feature fusion: (5)$$\begin{array}{c} F^{\prime }=\sum_{j=1}^{K}{\tilde{\omega }}_{j}\cdot F_{j},\;{\tilde{\omega }}_{j}=\frac{\omega_{j}}{\sum_{k=1}^{K}\omega_{k}}. \end{array}$$

Incorporating this mapping relationship enables the creation of networks capable of accommodating input sequences of unlimited length while maintaining fixed-dimensional features in the results. On the one hand, the network exhibits minimal structural and parametric expansion as the input image sequence length *K* increases. On the other hand, it associates input images with their respective image features, effectively preserving the original image information alongside the significance of feature maps. Simultaneously, this module’s operation is straightforward and effectively manages computational complexity. In comparison to fusion techniques utilizing maximum or minimum values, this training approach offers superior stability and mitigates gradient truncation concerns. Unlike the use of the mean value for fusion, it takes into account the allocation of exposure information weights in input images, thereby averting bias issues stemming from inconsistent exposure information. Employing normalization for weight calculation, rather than simple summation [7], ensures the stability of weight values, preventing abrupt value range expansion as *K* increases. Hence, adopting a weighted approach represents the optimal method for feature fusion. We denote this fusion strategy as ’Exposure-Guided Feature Fusion’ (EGFF).

#### 3.2.2. Gated Context Aggregation Network

The primary objective of the context aggregation network (CAN) [32] is to aggregate context information at a deeper level without compromising spatial resolution in order to capture context information at various scales. In their research, they examined multiple convolutional network architectures and determined that the context aggregation network based on dilated convolutions has significant advantages in terms of approximate accuracy, speed, and convergence. Building upon their work, we have adopted a similar CAN architecture and made enhancements by incorporating a gate fusion structure to extract features at different levels, which we refer to as GCAN, as shown in Figure 3.

The fundamental component of CAN is dilated convolutions; although dilated convolutions are widely acknowledged as effective, they can generate a phenomenon known as gridding artifacts [33]. To mitigate this issue, Wang et al. [34] proposed either introducing interactions between input units before dilated convolutions or appending a convolutional layer with a kernel size of 2r-1 after dilated convolutions. In this study, we have chosen to employ a similar approach by incorporating depthwise separable convolutions as a pre-convolution layer prior to dilated convolutions.

Depthwise separable convolutions typically consist of two stages: depthwise convolutions and pointwise convolutions (also referred to as 1 × 1 convolutions), which independently address spatial and cross-channel correlations in the input. In the depthwise convolution stage, each input channel undergoes convolution with a distinct set of filters to effectively capture spatial correlations within each channel. Subsequently, in the pointwise convolution stage, the output of the depthwise convolution is convolved with 1 × 1 filters, taking into account cross-channel correlations. These shared filters enable each output feature to be a function of all input channels, thereby enhancing the dependency of the output on the input units.

In computer vision, the integration of features from diverse layers can foster both low-level and high-level tasks. With this in mind, we have developed a structure, G, for gated fusion. Initially, this structure extracts feature mappings from an array of levels *F*, subsequently feeding them into a fusion block. The fusion block’s role is to compute weights for three distinct levels, each corresponding to a unique feature level. This design aims to achieve gated fusion weights that can efficiently aggregate information across various levels.
(6)$$\begin{array}{c} \left(W_{l},W_{m},W_{h}\right)=G\left(F_{l},F_{m},F_{h}\right),\\ F_{o}=W_{l}*F_{l}+W_{m}*F_{m}+W_{h}*F_{h}, \end{array}$$

In our proposed GCAN module, $F_{l}$, $F_{m}$, and $F_{h}$ denote the feature information at the low, middle, and high layers, correspondingly. Concurrently, $W_{l}$, $W_{m}$, and $W_{h}$ serve as the output weights of the gated structure. $F_{o}$ signifies not just the weight post-information aggregation, but also embodies the outcome of the whole GCAN module.

Simultaneously, the reconstruction network was enhanced by incorporating a global residual structure, thereby transforming the feature fusion process into a residual learning process, which facilitates network training. The inclusion of skip connections not only enables more comprehensive utilization of image information but also ensures that the fused image closely resembles the reference image, thereby preserving the color and brightness relationships within the image. More specifically, this design effectively reuses shallow features from the reference image, injecting spatial information containing these shallow features into middle-level features with a larger receptive field. This facilitates further integration and modulation of spatial and neighborhood information. This reuse of features can also be interpreted as learning the residual between the fused image features and the reference image features. Finally, a decoder structure symmetrical to the encoder is employed to maintain the original resolution and convert the feature maps back into the original image space, resulting in the successful fusion of multi-exposure images.

### 3.3. Loss Function

The Multiple Exposure Image Fusion Structural Similarity (MEF-SSIM) [19] evaluates the quality of image fusion by measuring the local similarity between the input image sequence and the fused image. This evaluation metric not only inherits the excellent mathematical properties of Structural Similarity Index (SSIM) but also aligns with people’s subjective perception of image quality. It also satisfies the requirement of unsupervised learning without the need for labels, making it widely applicable. However, this operator is only applicable to brightness information, which weakens the color information of the image to some extent. Based on this, Ma [8] proposed MEF-SSIMc to extend it to the color space. MEF-SSIMc is an evaluation index for multiple exposure image fusion that contains rich texture, color, and spatial structural information. The representation of MEF-SSIMc is: (7)$$\begin{array}{c} Q\left(\left\{L_{k}\right\},F\right)=\frac{1}{M}\sum_{j=1}^{M}\frac{\left(2\mu_{\hat{L}}\mu_{F}+C_{1}\right)\left(2\sigma_{\hat{L}F}+C_{2}\right)}{\left(\mu_{\hat{L}}^{2}+\mu_{F}^{2}+C_{1}\right)\left(\sigma_{\hat{L}}^{2}+\sigma_{F}^{2}+C_{2}\right)}. \end{array}$$

Among these variables, *M* denotes the number of patches, and ${\hat{\mu }}_{L}$ and *F* represent the expected mean intensity of image sequence patches and the fusion result, respectively. For brevity, please refer to reference X for the specific calculation formula of the mean intensity. $\mu_{\hat{L}}\mu_{F}$ signifies the covariance between ${\hat{\mu }}_{L}$ and *F*, whereas $C_{1}$ and $C_{2}$ are small constants introduced to prevent instability when the denominator approaches 0. Formula X is employed to compute the comprehensive quality measurement of the MEF-SSIMc indicator.

However, experiments have revealed that the use of MEF-SSIMc alone to formulate the loss function is considerably effective in preserving the main structural and color components of the image. Nonetheless, it also introduces blurring to the fused image, leading to a substantial loss of mid-to-high frequency details. Consequently, it becomes necessary to incorporate certain constraint terms into the loss function to ensure the preservation of mid-to-high frequency details in the image. To this end, we strive to introduce a gradient fidelity term into the loss function. The definition of the gradient fidelity term $E_{\nabla }$ is provided below: (8)$$\begin{array}{c} E_{\nabla }=\left.|\sum_{k=1}^{K}\tilde{\omega_{k}}\nabla L_{k}-\nabla F\right.{|}_{2}, \end{array}$$
where ∇ is the gradient operator, $\tilde{\omega_{k}}$ is the normalized weight. The specific calculation formula can be referred to in Equation (Equation 6). *F* represents the fused image. Our ultimate optimization goal is to simultaneously maximize MEF-SSIMc and minimize the gradient fidelity term. Maximizing MEF-SSIMc is used to optimize the general quality of the fused image, including spatial information, color information, exposure information, and mid-to-low frequency components. Minimizing $E_{\nabla }$ is used to optimize the details of the mid-to-high frequency components. Therefore, we can construct the following loss function: (9)$$\begin{array}{c} L=\alpha \cdot E_{\nabla }-E_{MEF-SSIM_{c}}+1. \end{array}$$

In this paper, the balance parameter α is set to 0.9.

Throughout the experiments conducted on the window size of MEF-SSIMc, a significant improvement in the overall fusion effect was observed as the window size was increased from 8 to 19. However, surpassing a window size of 35 did not result in the anticipated performance improvement, but rather led to an increase in computational time. As a result, for this paper, the window size for MEF-SSIMc has been set to 34.

Consequently, the construction of the loss function involves only the input image sequence and the output of the algorithm, without any ground truth. Therefore, our network is trained in an unsupervised manner.

## 4. Experimental Results and Comparisons

In this section, we conducted experiments to validate the fusion performance of the proposed Ref-MEF. First, we provided a detailed description of the experimental setup. Second, we compared Ref-MEF with recent classical MEF methods through subjective evaluation and objective assessment. Finally, we performed a series of ablation experiments to assess the effectiveness of the core components.

### 4.1. Training

We collected a comprehensive dataset for Ref-MEF. Initially, we gathered over 842 exposure sequences from six different sources [7,8,19,25,35,36]. Additionally, we used handheld devices and tripods to capture 30 sets of multi-exposure image sequences. We first eliminated sequences containing obvious object movement and retained only those successfully aligned using existing image registration algorithms [13,14,15]. After strict screening, a total of 794 static sequences were preserved. These sequences included rich HDR content, covering indoor and outdoor environments, static objects, and daytime and nighttime scenes. Their spatial resolution ranged from 0.2 to 20 million pixels, with exposure counts varying between three and nine times. Among them, 700 sequences were designated for Ref-MEF training, whereas the remaining 94 sequences were reserved for testing purposes.

During the training process, we adjusted the dimensions of the exposure sequences to three different resolutions: 128 s, 512 s, and 1024 s, representing low, medium, and high resolutions, respectively. Specifically, 128 s denotes resizing the shorter side to 128 while preserving the aspect ratio. We trained the entire network using the Adam optimizer for 100 epochs, with a learning rate of $10^{-4}$. The remaining parameters in the Adam optimizer were kept at their default values. The batch size was set to match the number of exposures in the current sequence. To strike a balance between runtime and performance, we configured the seven dilation rates of GCAN as follows: (2, 2, 2, 4, 4, 4, 1). Additionally, we set the channel count for all intermediate convolutional layers to 64.

### 4.2. Main Result

Our study involved a comparative analysis of Ref-MEF with twelve state-of-the-art MEF techniques. These encompass traditional spatial and transformation domain-based algorithms such as SPD-MEF [17], MEF-Opt [8], FMMEF [37], and GD [3]. Additionally, we scrutinized several advanced deep learning methodologies, including DeepFuse [7], MEFCNN [38], IFCNN [11], FusionDN [20], U2Fusion [21], MEF-GAN [9], MEF-Net [12], and Trans-MEF [30].

In deep learning methods, aside from MEF-Net [12] which can flexibly handle multiple inputs, the number of multiple exposure images processed by other networks is fixed. To ensure the consistency of the comparison, we adopted a nested model strategy, that is; first selecting two images to merge to obtain an intermediate result, then merging this result with other LDR images, and repeating this process until the final result is obtained, thus ensuring that the number of inputs for all networks is the same.

Partial results of our algorithm’s execution are depicted in Figure 4.

#### 4.2.1. Qualitative Comparison

The scene depicted in Figure 5 captures the exterior of a 7-Eleven convenience store during the night, encompassing both nocturnal and illuminative conditions. Under conditions of extreme exposure, the integration of information within the underexposed and overexposed regions becomes exceedingly challenging, presenting a formidable task for the algorithm. In contrast to both DeepFuse and our proposed method, alternative approaches generally prove inadequate in effectively mitigating highlights within the vicinity of street lamps. Within SPD-MEF, MEF-Opt, and MEF-Net, substantial errors manifest in the region illuminated by street lamps. Despite these challenges, our algorithm adeptly magnifies the light bulb in the area illuminated by street lamps. Regarding the section depicting the convenience store, SPD-MEF, MEF-Opt, MEF-Net, and TransMEF tend to manifest color deviations or insufficient brightness, resulting in a substantial loss of details. IFCNN exhibits an overall diminished brightness, and the fusion outcomes of DeepFuse and FusionDN demonstrate reduced clarity. Upon meticulous examination of the results, our algorithm excels in preserving the nuanced texture details of ground tiles and the internal information of the convenience store. Consequently, in the context of the current intricate scene examination, our algorithm emerges as particularly proficient in both information preservation and the suppression of highlights.

Figure 6’s environment, atop a shopping mall, presents a wide exposure range challenge for fusion algorithms. MEF-Opt, FMMEF, MEF-CNN, MEF-Net, and Trans-MEF suffer from pronounced ghosting artifacts, particularly near windows with direct sunlight, impacting the overall visual quality. Additionally, SPD-MEF produces an overly bright and sharp image, whereas GD and DeepFuse yield overall dark results. IFCNN shows significant color deviation, with a predominant greenish tint. In contrast, FusionDN, U2Fusion, and Ref-MEF offer superior subjective visual effects. Notably, Ref-MEF adeptly handles multiple exposure challenges, such as ghosting, over-sharpening, and color deviation, providing balanced visuals in the challenging top-floor indoor setting, positioning it among the best in this context.

In Figure 7’s coastal scene, comprising coastline, sunset, clouds, and waves, several fusion algorithms face hurdles. U2Fusion, MEF-GAN, and TransMEF exhibit notable errors; MEF-GAN shows clear color deviation, and both MEF-GAN and TransMEF significantly lose detail, obscuring the coastline. FMMEF, DeepFuse, IFCNN, and FusionDN produce overall dark images with limited dynamic range. MEF-Net and MEF-CNN struggle with texture detail degradation, especially in distant clouds and waves. For SPD-MEF, MEF-Opt, GD, and Re-MEF, the lack of Ground Truth hampers further subjective evaluation, though they show a high dynamic range.

Figure 8 depicts the “ColorChecker” scenario, wherein the original scene comprises two color palettes separated by a barrier for light control. One side of the barrier is exposed to extremely dim light, whereas the other side receives direct illumination. The distribution of lighting information in the scene is highly variable, presenting a formidable challenge for the MEF algorithms in recovering authentic scene details. Figure 8 showcases the outcomes of various algorithms, revealing substantial lens flare errors in SPD-MEF. IFCNN, FusionDN, U2Fusion, and MFE-GAN demonstrate perceptible shortcomings in restoring and preserving color palette details within shaded regions. The fusion results of GD, DeepFuse, IFCNN, FusionDN, U2Fusion, MEF-GAN, and Trans-MEF all manifest pronounced shifts in white balance, significantly diverging from the genuine color information of the scene. MEF-Opt, FMMEF, and MEFNet produce results more closely aligned with the authentic color tones of the scene. Regrettably, MEF-Opt yields somewhat blurry images. Given the current scenario, we posit that the outcomes from FMMEF, MEFNet, and our approach most faithfully represent the true scene.

Figure 9 portrays the nocturnal scene outside Waffle House, featuring a stationary red car that serves as a focal point in the current context. The comparison of fusion results is focused on these key elements. In the fusion outcomes, FMMEF manifests conspicuous errors. The majority of algorithms encounter challenges in rendering visible details, particularly in the bicycle’s front window and the driver’s cabin, with U2Fusion and MEF-GAN verging on operational failure. Although IFCNN adeptly retains finer details, the overall image significantly diverges from human perceptions of a nocturnal setting, exhibiting excessive brightness and suffering from overexposure in the Waffle House window area. In this specific scenario, we contend that our algorithm, when juxtaposed with alternative methods, meticulously preserves details and aligns most closely with the human perception of a nighttime scene.

Figure 10 illustrates the garden scene. In the current context, the overall outputs of IFCNN, U2Fusion, MEF-GAN, and TransMEF appear dim, revealing noticeable shortcomings in detail preservation. Notably, MEF-GAN introduces artificial clouds in the sky to enhance natural color transitions; however, a distinct boundary between the white clouds and the blue sky should be maintained. MEF-Net exhibits pronounced ghosting effects, particularly evident in color deviations surrounding the leaves on the right side of the image. In this scenario, the subjective representations of alternative algorithms are generally satisfactory, primarily reflecting subjective selection differences rather than significant errors. Our algorithm, in this particular scenario, effectively retains both detailed and color information.

In summary, our algorithm exhibits exceptional robustness in managing intricate scenes, particularly in the context of sequence fusion. The majority of learning-based algorithms typically embrace a design that limits the input to a pair of images, achieved through nested models for sequence image fusion. However, this approach falls short of fully capitalizing on exposure information across the entire sequence. It is noteworthy that, in certain relatively uncomplicated scenarios, even when errors are infrequent, there may be variations in subjective assessment results among individuals. Nevertheless, our algorithm delves into the correlations within the entire exposure sequence, meticulously addressing aspects such as color accuracy, image details, and overall effectiveness, with the aim of attaining satisfactory outcomes in diverse situations.

#### 4.2.2. Quantitative Comparison

Numerous methods primarily utilize MEF-SSIM [19] as an evaluation metric in MEF research. This metric primarily assesses the structural similarity between the fused image and the inputted image sequence, correlating with intuitive human cognition. However, this method predominantly considers brightness details, partially disregarding color information. To rectify this limitation, Ma [8] expanded MEF-SSIM into a color space, generating MEF-SSIMc. This updated evaluation metric simultaneously considers the image’s color, texture, and spatial structure details, offering a more thorough assessment of the effect of multiple exposure image fusion.

Table 1 shows the numerical comparison results of MEF-SSIM and MEF-SSIMc. Among all the compared methods, MEF-Opt [8] performed the best, which is not surprising, because its algorithm is designed to optimize these two indicators in the global image space. Compared with other methods, Ref-MEF is close to the best in MEF-SSIMc and ranks third in MEF-SSIM, which fully proves the effectiveness of our Ref-MEF network training.

Furthermore, given that the MEF task remains within the scope of image fusion, several prevalent image quality assessment metrics are appropriate for evaluating its outcomes. Specifically, image fusion results should maximize detail retention and minimize artifact presence for enhanced visual appeal. Consequently, to evaluate the MEF method comprehensively, we have incorporated an expanded set of standard image quality assessment metrics. As illustrated in Table 2, we utilized 13 standard metrics to perform a thorough analysis of MEF fusion results, focusing on three domains: information theory-based, image feature-based, and human perception-inspired.

Table 3 demonstrates the calculated results derived from our chosen evaluation metrics. With respect to the image feature-based metrics, the Ref-MEF method exhibits substantial superiority. Simultaneously, in measurements grounded on information theory and human visual perception, it ascends to the benchmark level of contemporary algorithms. We apply an average to the rankings corresponding to each method across varied metrics, subsequently organizing them in accordance with their scores. These findings serve to confirm that our methodology secures a leading position, thereby authenticating its preeminent stature in the realm of objective evaluation metrics.

#### 4.2.3. Running Time Comparison

In order to compare the computational costs of the MEF algorithm, we demonstrate the running time with varying numbers of exposure shots, as depicted in Figure 11. These times are the average fusion times under a scenario with a fixed spatial resolution of 512 s (resizing the shorter side to 512 while preserving the aspect ratio). Among these algorithms, MEF-Opt [8] has the longest running time, which is not surprising as it employs iterative methods to perform global image optimization to satisfy the MEF-SSIM [19] criterion. Compared to other learning-based methods, Ref-MEF does not require more model layers due to its unique network structure when the number of exposure shots increases, and therefore the running time does not change much. However, other deep learning methods show a linear growth trend when the number of exposure shots increases. Although the algorithm of MEF-Net [12] is slightly faster than ours, our algorithm significantly outperforms MEF-Net in both objective and subjective assessments of the fusion results.

### 4.3. Ablation Experiments

To evaluate the significance of each component in Ref-MEF, we conducted an ablation analysis to explore the impact of different components on network performance. Special attention was directed towards the REC module, EGFF module, separable convolution, and gate fusion block in the GCAN, as well as the gradient fidelity term in the loss function. The influence of each component was systematically assessed by individually removing them. Upon removal of the REC module, we substitute the original structure with conventional convolution. For the exclusion of the EGFF module, a strategy akin to the weighted averaging of IFCNN [11] was employed to compute features. The GCAN was substituted with the CAN baseline network without separable convolution and gate fusion structures. In the discourse on the loss function, the gradient fidelity term was omitted, and only MEF-SSIMc was utilized for unsupervised training. Detailed configurations and experimental metric values can be found in Table 4.

Beyond the gradient fidelity term in the loss function, the removal of these components during the ablation process resulted in varying degrees of suppression on the performance of MEF-SSIM [19] and MEF-SSIMc [8]. It is crucial to note that the exclusion of the gradient fidelity term led to an improvement in the metrics of MEF-SSIM [19] and MEFSSIMc [8]. Nevertheless, we recommend retaining the gradient fidelity term due to its pivotal role in preserving texture and high-frequency information in images.

In Figure 12, We showcase fusion outcomes related to scenes involving books, achieved by adjusting various balance factors, represented as α. Upon closer inspection of the image text details, it is observed that in the absence of the fidelity term for gradients (α=0), even though MEF-SSIMc based on the image structure reaches its maximum, the image details remain incomplete, resulting in inadequate clarity. Experimental findings reveal that as α increases to 0.7, there is a notable enhancement in image clarity, as evidenced by metrics such as average gradient (AG) and edge intensity (EI), describing image features. A further increment of α to 0.9 leads to a stabilization of image clarity. However, metrics related to image features, such as AG and EI, exhibit limited improvement at this stage. To achieve a balance between image features and the preservation of the main image structure, the balance factor α is set to 0.9. This decision is made with performance considerations in mind, aiming to retain texture and details while maximizing the maintenance of the image’s primary structure.

It is noteworthy that, following the incorporation of the REC module, a substantial improvement in objective evaluation metrics has been achieved, evidenced by an almost 10% increase in MEF-SSIM and MEF-SSIMc on the test set. Furthermore, after the introduction of the REC module, notable enhancements in color deviation and scene tone consistency have been observed in the fusion results, as depicted in Figure 13. The REC module, utilizing an attention mechanism, thoroughly explores the color and exposure information within the exposure sequence, playing a pivotal role in providing essential corrective enhancements to features of other LDR images.

Utilizing separated convolution in CAN significantly strengthens the dependency of the dilated convolution input unit, aiding in the suppression of grid artifacts [34]. We contrast this approach with the classical CAN network [32] and present two examples. As depicted in Figure 14, in parts (a) near the leaves and railing, and (b) near the clock tower and leaves, grid artifacts and color deviations are evident at the edges of objects or texture regions. Nevertheless, employing separated convolution effectively resolves these issues while preserving the original fidelity of the image, thus demonstrating the efficacy of our design.

## 5. Conclusions

We propose a flexible multi-exposure image fusion method for static scenes, named Ref-MEF, which can accept an uncertain number of multi-exposure image inputs. To correct the color difference and inconsistency in hue in the fusion results, we have constructed a module called reference-guided exposure correction (REC). Using the exposure-guided feature fusion (EGFF) module, we have realized adaptive weighting to construct a fixed feature dimension. In addition, we use the reference-guided exposure correction (GCAN) to aggregate multi-scale contexts, reducing grid artifacts. In the loss function, we introduce a gradient fidelity term to ensure the completeness of detail information and high-frequency information in the fusion results. Our method shows excellent performance both qualitatively and quantitatively and is more flexible in handling an increasing number of multi-exposure inputs.

## Figures and Tables

**Figure 1 entropy-26-00139-f001:**
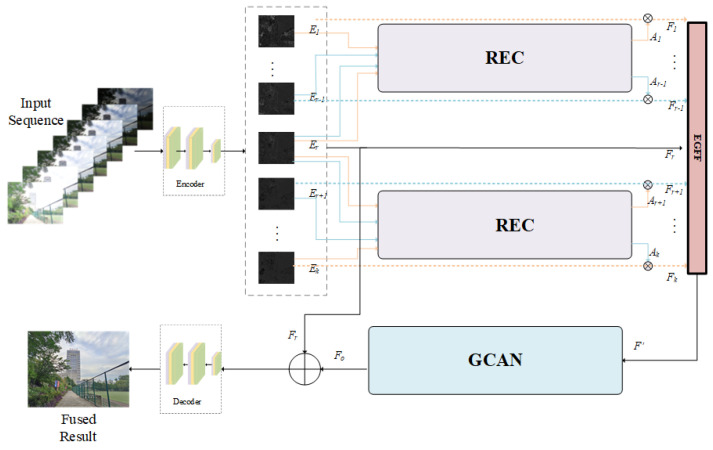
The overall network architecture of Ref-MEF, as proposed, follows a basic autoencoder framework. It consists of an encoder section with three convolutional layers. The reference-guided exposure correction (REC) module is responsible for feature correction, exposure-guided feature fusion (EGFF) handles feature fusion, and gated context aggregation network (GCAN) along with global residual connections are responsible for feature reconstruction. The decoder section is tasked with restoring the image to the RGB image space.

**Figure 2 entropy-26-00139-f002:**
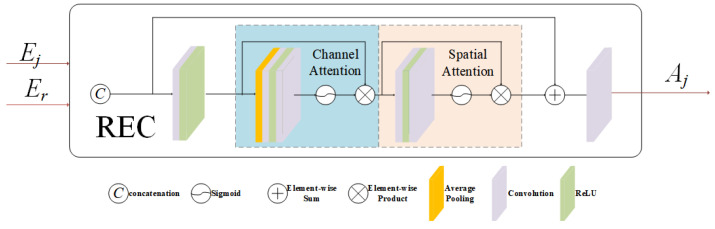
Reference-guided exposure correction module (REC).

**Figure 3 entropy-26-00139-f003:**
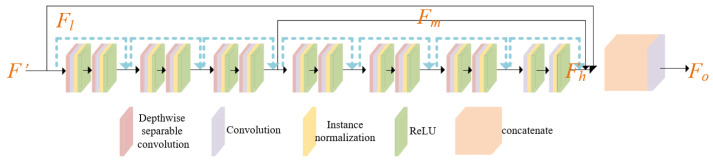
The architecture of gated context aggregation network (GCAN).

**Figure 4 entropy-26-00139-f004:**
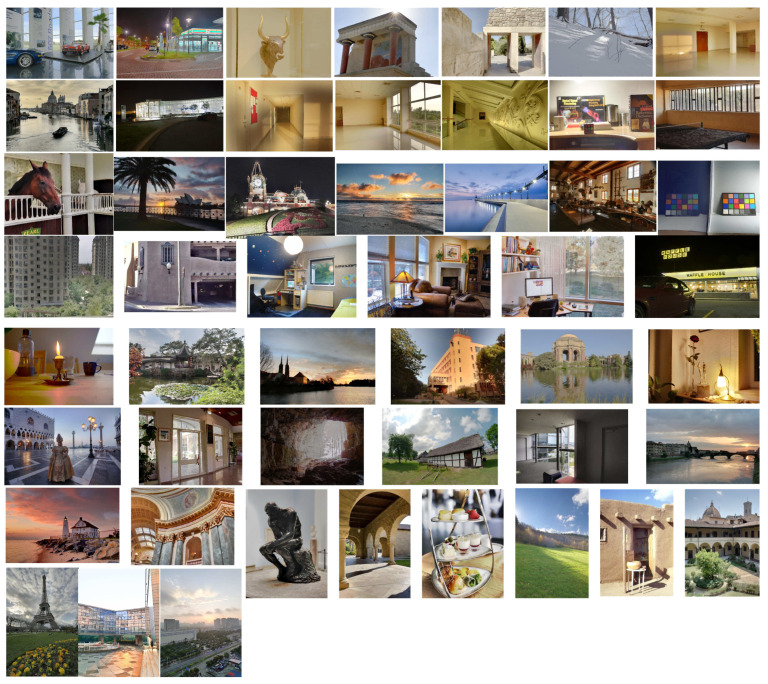
Results from partial execution of the Ref-MEF method.

**Figure 5 entropy-26-00139-f005:**
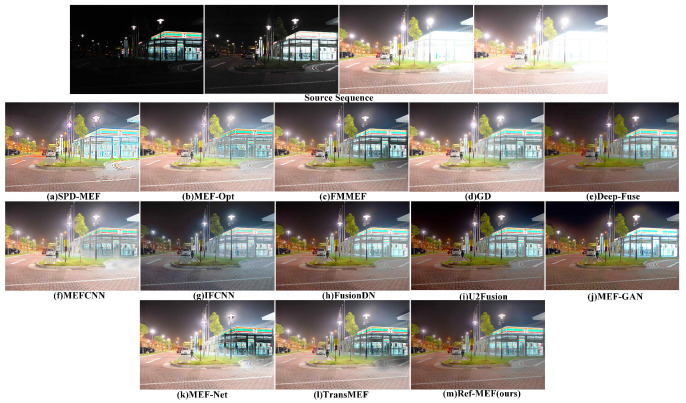
The qualitative performance comparison on the “SevenElevenNight” image sequence.

**Figure 6 entropy-26-00139-f006:**
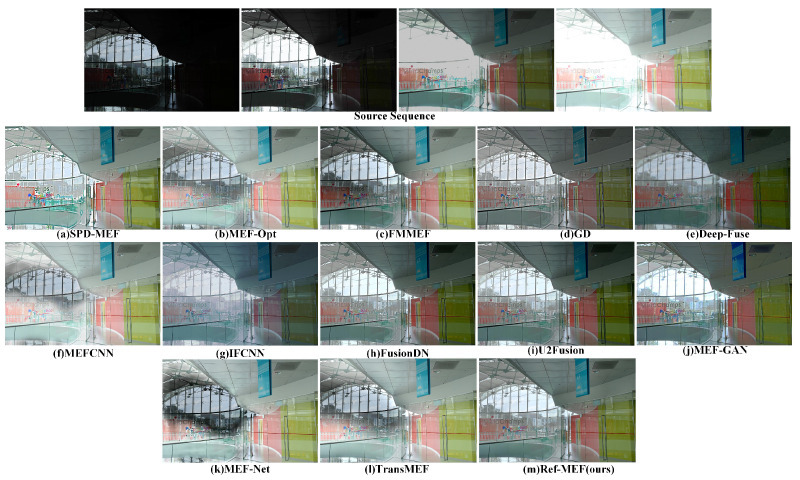
The qualitative performance comparison on the “Preschool” image sequence.

**Figure 7 entropy-26-00139-f007:**
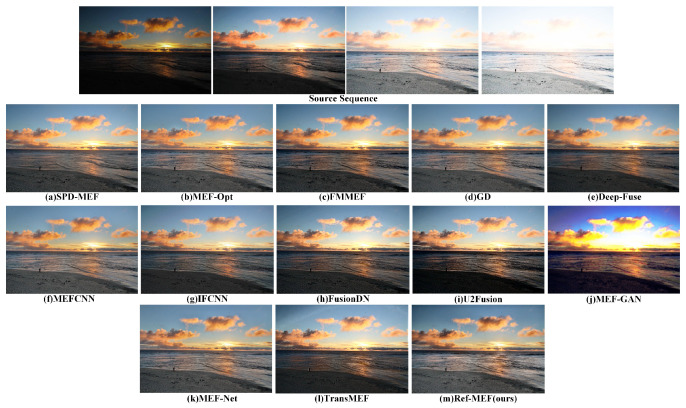
The qualitative performance comparison on the “Sky” image sequence.

**Figure 8 entropy-26-00139-f008:**
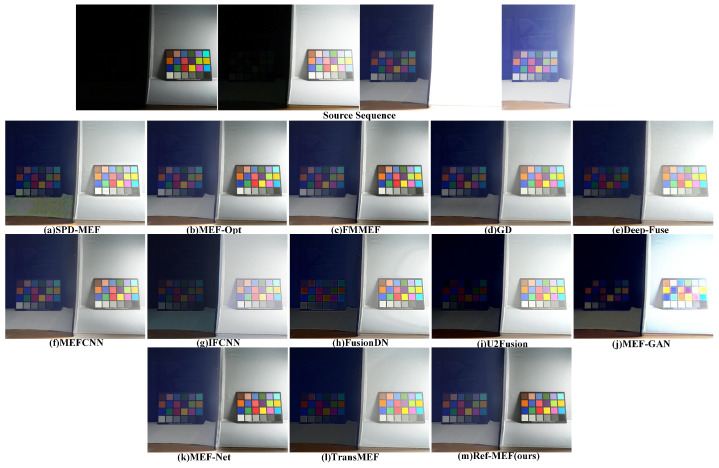
The qualitative performance comparison on the “ColorChecker” image sequence.

**Figure 9 entropy-26-00139-f009:**
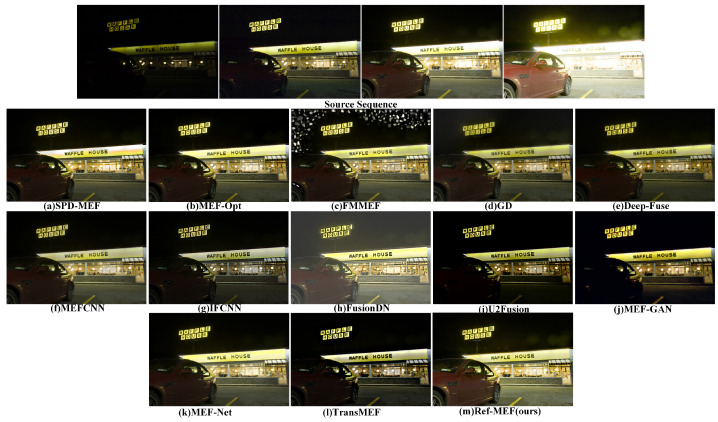
The qualitative performance comparison on the “WaffleHouse” image sequence.

**Figure 10 entropy-26-00139-f010:**
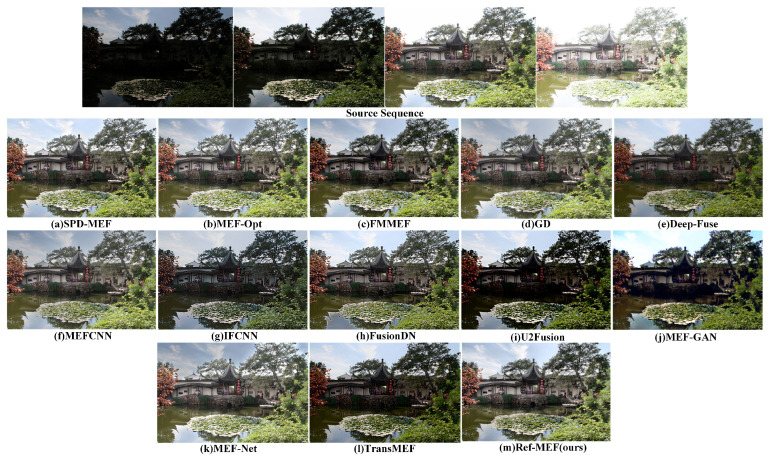
The qualitative performance comparison on the “ChineseGarden” image sequence.

**Figure 11 entropy-26-00139-f011:**
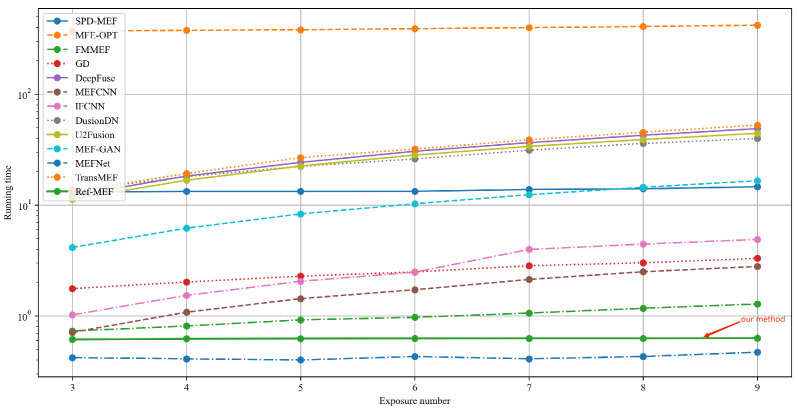
The change in the runtime of the MEF algorithms corresponds to an increase in the number of exposure lenses.

**Figure 12 entropy-26-00139-f012:**
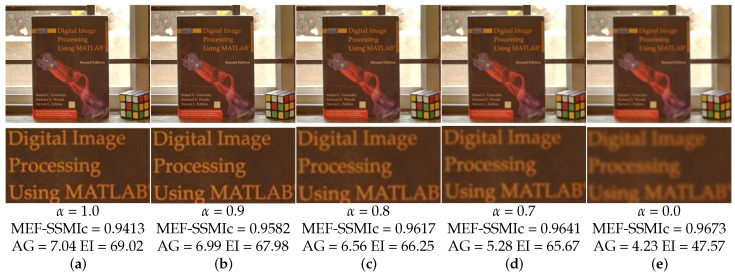
The fusion results obtained with different settings of the gradient fidelity term’s balance factor α in the “ICCV02” scenario. (**a**) When the balance factor is 1.0, (**b**) when the balance factor is 0.9, (**c**) when the balance factor is 0.8, (**d**) when the balance factor is 0.7, (**e**) when the gradient fidelity term is not used, the balance factor is 0.

**Figure 13 entropy-26-00139-f013:**
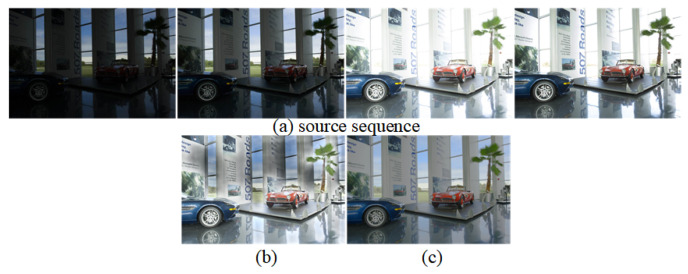
Subjective experimental results of the ablation of the REC module. In the first row, item (**a**) depicts the provided multi-exposure image sequence. The second row, item (**b**), presents outcomes in the absence of the REC module, whereas item (**c**) illustrates enhancements upon the module’s integration.

**Figure 14 entropy-26-00139-f014:**
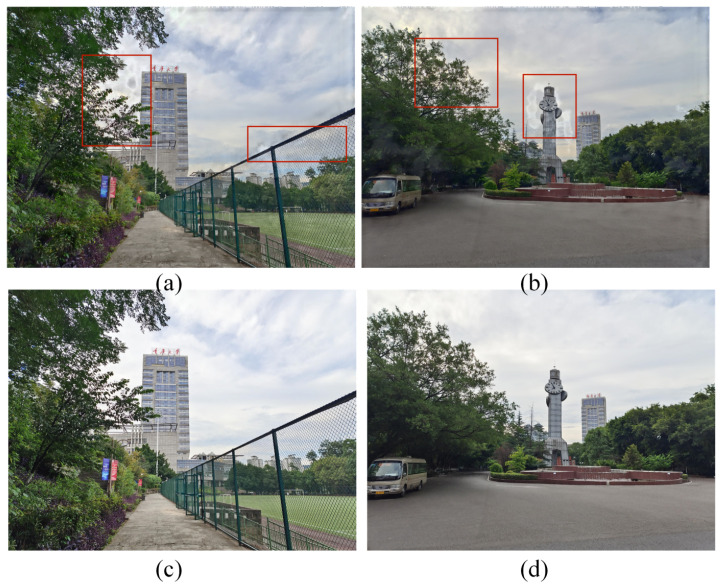
Subjective experimental results of the ablation of the separated convolution. The regions identified by the red boxes display notable grid artifacts. The first row (**a**,**b**) displays the results absent the use of separated convolution, whereas the effects of applying separated convolution are exhibited in the second row (**c**,**d**).

**Table 1 entropy-26-00139-t001:** Comparison of MEF-SSIM and MEF-SSIMc index values for various multi-exposure image fusion methods.

**Metrics**	SPD-MEF [17]	MFE-Opt [8]	FMMEF [37]	GD [3]	DeepFuse citedeepfuse	MEFCNN [38]	IFCNN [11]
MEF-SSIM	0.9382	0.9762 ^1^	0.9324	0.9645 ^2^	0.8968	0.9364	0.9432
MEF-SSIMc	0.9271	0.9775 ^1^	0.9403	0.9527 ^3^	0.862	0.9126	0.9237
**Metrics**	FusionDN [20]	U2Fusion [21]	MEF-GAN [9]	MEFNet [12]	TransMEF [30]	Ref-MEF	
MEF-SSIM	0.924	0.9304	0.7722	0.9139	0.8972	0.9496 ^3^	
MEF-SSIMc	0.9123	0.9203	0.7802	0.9026	0.9032	0.9582 ^2^	

x ^y^ signifies that the method’s value is ‘x’ under the present indicator, and its ranking stands at ‘y’.

**Table 2 entropy-26-00139-t002:** Evaluation metrics used in this paper.

Category	Name	Meaning	+/−
Information theory-based	EN [39]	Entropy	+
	CE [40]	Cross entropy	−
	TE [41]	Tsallis entropy	+
	PSNR [42]	Peak signal-to-noise ratio	+
	NMI [43]	Normal mutual information	+
Image feature-based	AG [44]	Average gradient	+
	EI [45]	Edge intensity	+
	SD [46]	Standard deviation	+
	SF [47]	Spatial frequency	+
	Q^*AB/F*^ [48]	Gradient-based fusion performance	+
Human perception-inspired	Q^*CB*^ [49]	Chen–Blum metric	+
	Q^*CV*^ [50]	Chen–Varshney metric	−
	VIF [51]	Visual information fidelity	+

“+” indicates better performance with larger values, and “−” indicates better performance with smaller values.

**Table 3 entropy-26-00139-t003:** Average results obtained using various evaluation metrics by different MEF methods on the test dataset.

Methods	EN	CE	TE	PNSR	NMI	AG	EI	SD	SF	Q ^*AB/F*^	Q ^*CB*^	Q ^*CV*^	VIF
**SPD-MEF ^6^**	7.1811	3.234	17,394	58.5365	0.6984	5.8798	54.6891	56.7475	20.7963	0.6376	0.4546	354.9691	0.774
**MFE-OPT ^5^**	7.2264	3.2354	**152,840**	**58.5998**	0.5926	5.7986	58.7073	51.5027	19.5281	0.6898	0.4627	729.1273	0.6959
**FMMEF ^4^**	**7.4264**	2.9075	53,255	57.6855	0.4809	5.69	53.4626	53.508	18.9879	0.7006	0.4558	621.063	0.9041
**GD ^10^**	7.2257	3.7746	30,000	56.6983	0.5465	5.4076	53.4681	55.8072	17.8313	0.6749	0.4305	336.0762	0.8511
**DeepFuse ^12^**	6.8395	3.1395	93,099	57.9744	0.7403	3.4418	35.2964	47.8109	10.6415	0.3866	0.391	361.694	0.5178
**MEFCNN ^11^**	7.3061	**2.6457**	101,951	54.2667	0.5974	4.9264	51.0512	55.7865	17.1608	0.6667	0.4297	750.0043	0.7355
**IFCNN ^7^**	7.153	3.3971	47,282	55.3554	0.7796	6.1824	62.1918	51.5826	21.0011	0.5919	0.41	**238.3928**	0.7146
**FusionDN ^2^**	7.4243	2.9392	9673	54.9748	0.7383	6.9693	**69.3412**	**67.7641**	21.6542	0.536	0.4355	322.5755	**0.9505**
**U2Fusion ^9^**	6.6785	3.018	20,326	56.0697	0.7639	5.4728	59.3588	65.1615	18.5468	0.5354	0.4159	242.4821	0.8281
**MEF-GAN ^13^**	6.9109	2.773	21,360	54.857	0.5699	4.5945	48.5215	63.734	13.9918	0.2829	0.3822	618.3198	0.5859
**MEFNet ^3^**	7.3035	3.059	83,157	57.7134	0.6077	5.8818	62.828	58.5405	19.7768	0.6767	**0.4863**	622.814	0.8342
**TransMEF ^8^**	7.2123	3.0568	18,812	56.9614	**0.8031**	5.5395	54.5106	62.901	18.3247	0.5705	0.4197	281.2867	0.8175
**Ref-MEF ^1^**	7.4247 ^2^	2.93 ^4^	97,332 ^3^	58.5548 ^2^	0.7606 ^4^	**6.9852** ^1^	67.9798 ^2^	67.103 ^2^	**21.7384** ^1^	**0.7013** ^1^	0.4474 ^5^	292.275 ^4^	0.8681 ^3^

x ^y^ represents that x is ranked as the yth in the column. Here, x can refer to either a method or a specific evaluation metric. The data highlighted in bold indicates that the current method outperforms all contrastive methods within the current evaluation Metrics.

**Table 4 entropy-26-00139-t004:** Specific ablation settings and training configurations were implemented for each component, which demonstrate the optimal performance achieved by the combination of our designed modules.

**component**	**configuration1**	**configuration2**	**configuration3**	**configuration4**	**configuration5**	**configuration6**
REC		✓	✓	✓	✓	✓
EGFF	✓		✓	✓	✓	✓
Spe Conv	✓	✓		✓	✓	✓
Gated Fusion	✓	✓	✓		✓	✓
Loss with $E_{\nabla }$	✓	✓	✓	✓		✓
MEF-SSIM	0.8633	0.8479	0.8555	0.8958	**0.9588**	0.9496
MEF-SSIMc	0.8711	0.8578	0.8632	0.9039	**0.9673**	0.9582

## Data Availability

Data are contained within the article.

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
