# Peer review of "Ref-MEF: Reference-Guided Flexible Gated Image Reconstruction Network for Multi-Exposure Image Fusion"

_entropy, 2024, doi:10.3390/e26020139_

Round 1

Reviewer 1 Report

Comments and Suggestions for Authors

The paper proposes a MEF method based on deep learning that can accept an varying number of multi-exposure image inputs. This is well-structured and well-written paper. The Authors clearly present their concept and validate its efficiency by comparing their results with many existing methods.

The Authors propose a Reference-Guided Exposure Correction module that constructs a fixed feature dimension from a number of image inputs, that is then used by a reconstruction network (exposure-guided feature fusion and context aggregation network).

The RAC module requires the reference image from the exposure sequence. The Authors select it through the peak-to-average ratio. In my opinion, the influence of reference image selections might be more extensively discussed in the paper.

There is a lot of research on the topic and the Authors refer to many of them, but there are papers that should also be included in the Reference section and maybe used for comparison.

Zhao, H., Zheng, J., Shang, X. et al. Coarse-to-fine multi-scale attention-guided network for multi-exposure image fusion. Vis Comput (2023). https://doi.org/10.1007/s00371-023-02880-4

Yang, Z., Chen, Y., Le, Z. et al. GANFuse: a novel multi-exposure image fusion method based on generative adversarial networks. Neural Comput & Applic 33, 6133–6145 (2021). https://doi.org/10.1007/s00521-020-05387-4

K. Wu, J. Chen, Y. Yu and J. Ma, "ACE-MEF: Adaptive Clarity Evaluation-Guided Network With Illumination Correction for Multi-Exposure Image Fusion," in IEEE Transactions on Multimedia, vol. 25, pp. 8103-8118, 2023, doi: 10.1109/TMM.2022.3233299.

Author Response

Title: Reference-Guided Flexible Gated Image Reconstruction Network for Multi-exposure Image Fusion

Journal title: Entropy

Manuscript ID: entropy-2770703

Authors: Yuhui Huang, Shangbo Zhou, Yufen Xu, Yijia Chen, Kai Cao

Dear Reviewer,
We sincerely appreciate your thorough review of our manuscript and the valuable feedback provided. Your professional insights are highly regarded, and we have diligently considered and incorporated the suggested revisions into the paper.

Point1:
The REC module requires the reference image from the exposure sequence. The Authors select it through the peak-to-average ratio. In my opinion, the influence of reference image selections might be more extensively discussed in the paper.
Reply:
We provide a more detailed exposition of the rationale behind selecting peak-to-average ratio(PAR) as the reference image. Regarding the PAR of the histogram, we can distinctly observe instances of exposure distortion. The reference image should steer clear of exposure distortion, be it overexposure or underexposure, to preempt the introduction of inaccurate gains to the non-reference image.

We provide a more detailed exposition of the rationale behind selecting peak-to-average ratio(PAR) as the reference image. Regarding the PAR of the histogram, we can distinctly observe instances of exposure distortion. The reference image should steer clear of exposure distortion, be it overexposure or underexposure, to preempt the introduction of inaccurate gains to the non-reference image.

It is pivotal for the REC module to opt for a reference image devoid of exposure errors, a consideration far more critical than choosing an image with elevated image quality metrics, such as contrast or peak signal-to-noise ratio(PSNR). Despite our exploration of alternative methods, such as the maximum contrast approach and techniques grounded in image quality assessment, the chosen reference images remained largely consistent in most cases. Nevertheless, these methods may sporadically lead to errors, such as selecting an image with overexposed regions solely due to its high contrast or peak signal-to-noise ratio. We aim to avoid incorporating such images as reference images.

The selection of reference images is not the primary focus of our discourse. Our emphasis lies in ensuring that the chosen images are free from errors, a concern that surpasses the pursuit of optimality.

Point2:
There is a lot of research on the topic and the Authors refer to many of them, but there are papers that should also be included in the Reference section and maybe used for comparison.
Relpy:

The relevant papers has been incorporated into the references, duly acknowledging their impact on our work. However, regrettably, certain methods cited in the literature you provided lack public disclosure of their source code. Consequently, we are unable to perform a unified test, and consequently, these methods have not been included in our comparative analysis.

Our comparative methodology spans several domains exhibiting superior performance in the current landscape, encompassing conventional approaches, CNN-based techniques, GAN-based methodologies, and Transformer-based models, among others. We posit that such a comprehensive comparison framework will adequately evaluate the efficacy of our proposed method and facilitate comparisons with pertinent research across diverse domains.

We would like to express our gratitude for your guidance and constructive suggestions, which have significantly contributed to the improvement of our paper. We look forward to your reevaluation of our work and hope that our revisions meet your expectations.

Should you have any further questions or suggestions, please feel free to let us know.

Sincerely yours,

Shangbo Zhou on behalf of the authors.

Corresponding author: Shangbo Zhou.

Email: shbzhou@cqu.edu.cn

Reviewer 2 Report

Comments and Suggestions for Authors This paper proposes a multi-exposure image fusion method called Ref-MEF, which is a reference-guided flexible gated image reconstruction network. Both quantitative and qualitative results show the effectiveness of the proposed method.
I have some concerns listed below:

1. It would be nice to summarize the main contributions point by point at the end of the introduction part.

2. There lacks some recently MEF quality evaluation works, such as Quality assessment of multi-exposure image fusion by synthesizing local and global intermediate references.

3. The ablation experiments seem to be incomplete. It is suggested to remove each component and test the other parts.

4. Why not use some public datasets? Please clarify this.

5. Cross dataset validation is important for deep learning based models.

6. In Figure 1, there lack explanations/full names for each proposed module in the title.

7. In Figure 2, there also lack the explanations of the symbols in the title. Please check all the figures and tables to make them self-explanatory. Comments on the Quality of English Language

N/A

Author Response

Title: Reference-Guided Flexible Gated Image Reconstruction Network for Multi-exposure Image Fusion

Journal title: Entropy

Manuscript ID: entropy-2770703

Authors: Yuhui Huang, Shangbo Zhou, Yufen Xu, Yijia Chen, Kai Cao

Dear Reviewer,
We sincerely appreciate your thorough review of our manuscript and the valuable feedback provided. Your professional insights are highly regarded, and we have diligently considered and incorporated the suggested revisions into the paper.

**Addressing Key Concerns:**

Point1:

It would be nice to summarize the main contributions point by point at the end of the introduction part.
Reply:
We express our sincere gratitude for the invaluable feedback you have provided. Your suggestions have been thoroughly deliberated upon and incorporated into our work. We have meticulously outlined our primary contributions in a point-by-point fashion within the introduction section of the paper. This refinement significantly contributes to the overall readability of the article, and we are confident that these enhancements will better align with the readers' expectations. Once again, we extend our appreciation for your constructive suggestions, which play a pivotal role in advancing our research.

Point2:
There lacks some recently MEF quality evaluation works, such as Quality assessment of multi-exposure image fusion by synthesizing local and global intermediate references.

Reply:
Thank you sincerely for your valuable suggestions. We have extensively investigated the field of MEF quality assessment and recognized its vast scope for research. Our study is dedicated to proposing solutions for MEF, particularly in scenarios involving uncertain input quantities.

Despite our examination of the latest developments in MEF quality assessment, these innovative methods have not gained widespread application. Presently, numerous MEF approaches predominantly rely on MEF-SSIM for evaluation, with some methods being constrained solely to the use of MEF-SSIM. In our research, we introduced MEF-SSIMc, an enhanced version of MEF-SSIM that comprehensively incorporates color, texture, and spatial structure information of the images. Furthermore, we investigated universal image fusion evaluation metrics, encompassing indicators based on information theory, image features, and human visual perception. We appreciate your guidance, and we are committed to refining and continuously enhancing our research endeavors.

Point3:
The ablation experiments seem to be incomplete. It is suggested to remove each component and test the other parts.
Reply:
Regarding the design of the ablation experiments you mentioned, our initial strategy entailed constructing a baseline network and evaluating experimental outcomes through the progressive addition of modules. We opted for this approach to highlight the positive contributions of each component and methodically demonstrate the enhancement in network performance.

Nevertheless, we fully comprehend your suggestion and acknowledge that adopting an approach involving the systematic removal of each component can comprehensively assess the impact of each element, thereby furnishing more robust and complete experimental evidence. In the revised paper, we conducted experiments using the method of individually removing each component and presented corresponding experimental results.

Should you have any additional suggestions or requirements, please do not hesitate to inform us.

Point4:

Why not use some public datasets? Please clarify this.
Reply:
Regarding the question you mentioned about why we did not use public datasets, we would like to explain. Our dataset comprises two primary components: one portion is derived from publicly available datasets on the internet (including six data sources), and the other portion is a dataset we developed internally. In the datasets acquired from the internet, we incorporated the widely utilized SCIE dataset. However, we observed that even in these extensively employed datasets, instances of camera shake exist, leading to pseudo-images in the fusion results.

To address this issue, we conducted additional registration processing. To more precisely tackle the image registration problem, we amalgamated data from multiple sources to construct our proprietary dataset. We selected currently esteemed registration algorithms for the preprocessing of our proposed algorithm. This decision is intended to ensure that our method attains superior outcomes in managing static multi-exposure image fusion and enhances the algorithm's robustness. We trust this elucidates your concerns, and if there are other areas requiring clarification, please do not hesitate to inform us. We appreciate your time and professional advice.

Point5:

Cross dataset validation is important for deep learning based models.

Reply:
To ensure the robustness and generalization ability of our model, we have given special attention to the process of cross-dataset validation. Our dataset comprises multi-exposure image fusion data from various sources, including the publicly available SCIE dataset and a small dataset we curated ourselves. To maintain data quality, we conducted rigorous screening and registration preprocessing.

During the training phase, we employed a random sampling approach to extract samples from diverse data sources, resizing exposure sequences to three resolutions: 128s, 512s, and 1024s, corresponding to low, medium, and high resolutions, respectively. This training methodology aids the model in acquiring a more comprehensive understanding of universal features, thereby enhancing its performance across different datasets without overfitting to specific source data.

We are confident that this cross-dataset validation strategy fortifies the reliability of our model, rendering it more adaptable to a variety of real-world application scenarios.

Point6:

In Figure 1, there lack explanations/full names for each proposed module in the title.

Reply:
Thank you for your guidance. I have added the full names of each proposed module in the caption section of Figure 1 to ensure that readers have a clearer understanding of the content of each module. If you have any other suggestions or modifications, please feel free to let me know. Thank you for your time and professional guidance.

Point7:

In Figure 2, there also lack the explanations of the symbols in the title. Please check all the figures and tables to make them self-explanatory.

Reply:
Thank you for your guidance. In the caption of Figure 2, we have included the complete names of the proposed modules. It is advisable not to delve too deeply into the module details, as the preceding text in Figure 2 has already explained the symbols and provided a brief overview of each module's workflow. The approach to generating REC weights is also available in Formula 2. We have meticulously examined all tables and figures, striving to enhance their self-explanatory nature. Please feel free to promptly share any further suggestions or modifications.

We would like to express our gratitude for your guidance and constructive suggestions, which have significantly contributed to the improvement of our paper. We look forward to your reevaluation of our work and hope that our revisions meet your expectations.

Should you have any further questions or suggestions, please feel free to let us know.

Sincerely yours,

Shangbo Zhou on behalf of the authors.

Corresponding author: Shangbo Zhou.

Email: shbzhou@cqu.edu.cn

Reviewer 3 Report

Comments and Suggestions for Authors

This paper proposed deep learning-based MEF methodologies through combining a reference-guided exposure correction (REC) module and gated context aggregation network (GCAN).

Above all, performance verification is lacking. Sample images are also not enough, and there are many fusion networks with excellent performance recently. Comparisons with the methods below are required. (Various experimental images are required)

1. Li, H.;Wu, X.J. DenseFuse: A Fusion Approach to Infrared and Visible Images. IEEE Trans. Image Process. 2019, 28, 2614–2623.

2. Qu, L.; Liu, S.; Wang, M.; Song, Z. TransMEF: A Transformer-Based Multi-Exposure Image Fusion Framework Using Self-Supervised Multi-Task Learning. Proc. AAAI Conf. Artif. Intell. 2022, 36, 2126–2134.

3. Xu, H.; Ma, J.; Jiang, J.; Guo, X.; Ling, H. U2Fusion: A Unified Unsupervised Image Fusion Network. IEEE Trans. Pattern Anal. Mach. Intell. 2022, 44, 502–518.

Therefore, it is difficult to discuss the effectiveness of the performance of the proposed method, and the score performance cannot be considered particularly good. This part can be confirmed through subjective image verification. In particular, I am not convinced with the results of the existing method for figure 4. Many existing methods (including method-based techniques) achieve significantly better synthesis performance just by using 2 exposed images.

Comments on the Quality of English Language

Minor editing of English language required

Author Response

Title: Reference-Guided Flexible Gated Image Reconstruction Network for Multi-exposure Image Fusion

Journal title: Entropy

Manuscript ID: entropy-2770703

Authors: Yuhui Huang, Shangbo Zhou, Yufen Xu, Yijia Chen, Kai Cao

Dear Reviewer,
We sincerely appreciate your thorough review of our manuscript and the valuable feedback provided. Your professional insights are highly regarded, and we have diligently considered and incorporated the suggested revisions into the paper.

**Addressing Key Concerns:**

1. The issue of insufficient sample quantity that you mentioned is a current challenge constraining the development of Multi-Exposure Image Fusion(MEF). We have extensively collected data from various sources, including our dataset comprising the widely used SCIE dataset (many algorithms solely use this dataset for training and adopt data augmentation). We merged data from six sources and added a small portion of self-made datasets (under strict pre-registration processing). We designed learning at multiple resolutions (including low, medium, and high resolutions).

Currently, the dataset indeed presents a crucial practical issue in the development of MEF. TransMEF creatively employs a large-scale natural dataset (COCO dataset) for self-supervised learning, encompassing brightness, texture, and semantic information. However, the COCO dataset was initially used for object detection and image segmentation, not an HDR dataset, so the HDR information learned from the COCO dataset still faces limitations. Nonetheless, their proposed approach is innovative.

2. Concerning the three papers you mentioned, DenseFuse is specifically designed for the fusion of infrared and visible images, which diverges from the objectives of our MEF task. Nevertheless, among the comparative methods we consulted, FusionDN also employs a densely connected network, explicitly tailored for MEF training.

As you noted, both TransMEF and U2Fusion are already incorporated into our method comparisons. Our selection criteria for comparative methods aim to encompass mainstream or classical approaches comprehensively. This includes traditional spatial and hybrid domain methods such as SPD-MEF, FMMEF, GD, as well as image-space global optimization methods like MEF-Opt.

In the realm of deep learning methods, we introduced DeepFuse, MEFCNN, IFCNN, FusionDN, U2Fusion, MEF-GAN, MEF-NET, and TransMEF. Notably, MEF-GAN adopts a GAN-based approach, while TransMEF employs a Transformer-based approach. Our objective is to carry out a comprehensive and meticulous evaluation of methods spanning different categories.

3. Regarding the performance validation section, we included subjective and objective comparisons. Our objective comparison used metrics such as MEF-SSIM (most widely used) and MEF-SSIMc, along with some general image fusion evaluation metrics, including those based on information theory, image features, and human visual perception. We conducted extensive validation on the test dataset.

In the subjective experiment section, we supplemented with additional samples of algorithmic execution results (now in Figure 4), presenting more subjective examples.

We apologize for not convincing you regarding the original Figure 4's scrutiny (in point 4 of the response, we attempted to analyze the reasons), but these are indeed the execution results obtained by our algorithm compared to other algorithms. To better showcase the performance of our algorithm, we replaced that set of results (now in Figure 5), selecting a group of challenging scenarios for fusion algorithms, including night scenes and high-exposure extreme situations, to further demonstrate the excellence of our algorithm.

4. Concerning the matter you raised about enhancing synthesis performance using only two images, numerous existing methods, such as U2Fusion and IFCNN, exhibit proficiency in producing commendable outcomes from two highly exposed images. Nevertheless, the potential for substantial loss of intricate details in extremely exposed images poses a challenge, rendering the recovery of such lost information arduous even with a well-designed methodology.

To tackle this issue, we propose a method for fusing sequences of multiple exposure images of the same scene, applicable to any number of exposure images. Through the utilization of exposure coverage, our aim is to maximize the restoration of scene information, enabling the generated images to preserve abundant details and information while attaining noteworthy advantages in both subjective and objective evaluation metrics. Currently, obtaining multiple exposure images of the same scene is relatively convenient; multiple frames can be captured simultaneously by adjusting exposure levels, and our algorithm adeptly handles multiple inputs.

Numerous deep learning methods are constrained by network scale settings and predominantly utilize stitching for inputting images, thereby directly determining the input scale. Additionally, they often concentrate on the fusion of overexposed and underexposed image pairs. These algorithms have not conducted fusion experiments for scenarios involving moderate exposure or poor exposure quality (excluding extreme exposure). Consequently, these algorithms inherently possess certain limitations. Our method was compared with them; for scenarios involving exposure sequences (more than two images), we consistently processed their algorithms through model nesting. We initiated the fusion with two images to obtain intermediate results, followed by the incorporation of the third image. This iterative process continued until the entire exposure image sequence was comprehensively covered.

We would like to express our gratitude for your guidance and constructive suggestions, which have significantly contributed to the improvement of our paper. We look forward to your reevaluation of our work and hope that our revisions meet your expectations.

Should you have any further questions or suggestions, please feel free to let us know.

Sincerely yours,

Shangbo Zhou on behalf of the authors.

Corresponding author: Shangbo Zhou.

Email: shbzhou@cqu.edu.cn

Round 2

Reviewer 2 Report

Comments and Suggestions for Authors

The authors seem to fail to answer most of my comments, including the point by point contributions, more clarify of MEF quality evaluation, more ablation study, and cross dataset evaluation.

Comments on the Quality of English Language

N/A

Author Response

Title: Reference-Guided Flexible Gated Image Reconstruction Network for Multi-exposure Image Fusion

Journal title: Entropy

Manuscript ID: entropy-2770703

Authors: Yuhui Huang, Shangbo Zhou, Yufen Xu, Yijia Chen, Kai Cao

Dear Reviewer,
We sincerely appreciate your thorough review of our manuscript and the valuable feedback provided. Your professional insights are highly regarded, and we have diligently considered and incorporated the suggested revisions into the paper.

1. In relation to the point by point contributions you raised, we have encapsulated our contributions at the conclusion of the introduction and provided annotations in the most recent version of the manuscript. Kindly consult the latest manuscript, and should you have additional feedback, please do not hesitate to inform us.

2. In reference to the quality assessment work on MEF mentioned earlier, we acknowledge that our paper did not delve extensively into this aspect. However, we deem it unnecessary. Within the literature pertaining to Multiple Exposure Image Fusion (MEF), a distinct category is dedicated to exploring the quality assessment of multiple exposure image fusion, as evidenced in references[1-3]. These studies aim to enhance the quality assessment methodologies for MEF methods, contributing to the development of a robust evaluation framework.Another category of methods focuses on addressing the core challenges of MEF itself, exemplified by TransMEF, U2Fusion, IFCNN, IID-MFE, among others. Interestingly, these articles do not extensively discuss the quality assessment of MEF. Instead, their emphasis lies on designing algorithms to achieve superior fusion outcomes. Many of these methods employ MEF-SSIM and MEF-SSIMc for image quality assessment, specifically tailored for MEF tasks and widely embraced within the community. Additionally, some algorithms in papers rely on general image evaluation metrics such as Peak Signal-to-Noise Ratio (PSNR), average gradient, and others.

Our paper's evaluation framework includes MEF-SSIM and MEF-SSIMc dedicated to MEF quality assessment. Furthermore, we incorporate thirteen general image evaluation metrics across three dimensions—information theory, image features, and human visual perception. We contend that these multidimensional metrics offer a comprehensive assessment of MEF tasks. Notably, these evaluation metrics are drawn from widely recognized image evaluation systems, obviating the need for extensive elaboration.

3. In reference to the additional ablative experiments you mentioned, your previous comments highlighted that our ablative experiments lacked comprehensiveness and suggested disaggregating each component for individual testing. Initially, we adopted a baseline network approach and incrementally integrated components to accentuate their positive contributions, thereby reflecting the progressive enhancement of network performance.

In the latest manuscript, we have embraced your suggestion by conducting separate ablative validations for each proposed module and providing annotations in the most recent version. The specific configurations for the latest ablative experiments are delineated in Figure 4. Apart from the enhancement in network metrics achieved by removing the gradient fidelity term from the loss function, the exclusion of other components has adversely affected network performance. We also elucidate why the removal of the gradient fidelity term correlates with improved metrics. Gradient fidelity plays a pivotal role in preserving textures and high-frequency information in images to a certain extent. Hence, we advocate for the optimal configuration represented by the last column in Figure 4, rather than the penultimate column. Additionally, we deliberate on the selection of the balance parameter for the gradient fidelity term in the loss function. The ablative experiments are extensively annotated in the pertinent section of the latest manuscript.

4. Regarding the cross-data validation issue you raised, we seek to provide clarification. Currently, there is a limited availability of publicly accessible datasets specifically tailored for Multiple Exposure Fusion (MEF), with an even more pronounced scarcity for datasets focused on sequence fusion. Many methods, including DeepFuse, IFCNN, MEF-GAN, MEFCNN, U2Fusion, typically rely on a single dataset for both training and testing.

Our present strategy aligns with the approach proposed by MEF-Net, aiming to amass a diverse collection of public datasets. We have extensively curated data from six distinct sources, incorporating a small fraction of custom datasets. These datasets undergo meticulous screening and registration processes to ensure their quality and applicability. In order to foster algorithm generalization, we judiciously distribute sequence images from each dataset source proportionally between our training and testing sets.

However, concerning the aforementioned cross-data validation issue, aside from the SCIE dataset, which boasts ample data to support model training and testing, other individual datasets suffer from an insufficiency of samples, rendering them unsuitable for our training and testing criteria. This underscores our future trajectory, emphasizing the provision of additional datasets to facilitate the advancement of static sequence multiple exposure image fusion algorithms.

We would like to express our gratitude for your guidance and constructive suggestions, which have significantly contributed to the improvement of our paper. We look forward to your reevaluation of our work and hope that our revisions meet your expectations.

Should you have any further questions or suggestions, please feel free to let us know.

Sincerely yours,

Shangbo Zhou on behalf of the authors.

Corresponding author: Shangbo Zhou.

Email: shbzhou@cqu.edu.cn

Reference
【1】Shi, J.; Li, H.; Zhong, C.; He, Z.; Ma, Y. BMEFIQA: Blind Quality Assessment of Multi-Exposure Fused Images Based on Several Characteristics. Entropy 2022, 24, 285. https://doi.org/10.3390/e24020285

【2】Yang, Jiachen et al. “Blind quality assessment of tone-mapped images using multi-exposure sequences.” J. Vis. Commun. Image Represent. 87 (2022): 103553.

【3】Cui, Y.; Chen, A.; Yang, B.; Zhang, S.; Wang, Y. Human Visual Perception-Based Multi-Exposure Fusion Image Quality Assessment. Symmetry 2019, 11, 1494. https://doi.org/10.3390/sym11121494

Reviewer 3 Report

Comments and Suggestions for Authors

The author's answer seems reasonable. Nevertheless, there are not enough sample images to make a clear subjective judgment. Please add experiment results for color checker, waffle house, and garden images.

Author Response

Title: Reference-Guided Flexible Gated Image Reconstruction Network for Multi-exposure Image Fusion

Journal title: Entropy

Manuscript ID: entropy-2770703

Authors: Yuhui Huang, Shangbo Zhou, Yufen Xu, Yijia Chen, Kai Cao

Dear Reviewer,
We sincerely appreciate your thorough review of our manuscript and the valuable feedback provided. Your professional insights are highly regarded, and we have diligently considered and incorporated the suggested revisions into the paper.

We have successfully conducted the supplementary subjective experiments as stipulated. In certain relatively straightforward scenarios where the algorithms undergo less rigorous testing, the distinctions among most algorithms may not be readily apparent. Such discrepancies tend to be more aligned with individual preferences and stylistic choices, resulting in varied evaluation outcomes.

Nevertheless, our algorithm meticulously explores correlations within the entire exposure sequence, addressing color accuracy, image details, and overall effects with precision. Our goal is to attain satisfactory results across diverse situations. In contrast, many other learning-based algorithms predominantly employ image pair fusion designs, involving a fixed input of two images and utilizing a model nesting method for sequence image fusion. While they may demonstrate an advantage in fusing extreme exposure image pairs (comprising one overexposed and one underexposed image), their efficacy diminishes when applied to image sequences. In such cases, these algorithms may fail to exploit the comprehensive exposure information of the entire image sequence, leading to errors.

We would like to express our gratitude for your guidance and constructive suggestions, which have significantly contributed to the improvement of our paper. We look forward to your reevaluation of our work and hope that our revisions meet your expectations.

Should you have any further questions or suggestions, please feel free to let us know.

Sincerely yours,

Shangbo Zhou on behalf of the authors.

Corresponding author: Shangbo Zhou.

Email: shbzhou@cqu.edu.cn

Round 3

Reviewer 2 Report

Comments and Suggestions for Authors

Thanks for the authors' comments. The ablation study shows with our the proposed loss achieves the best performance, which should be explained more.

Comments on the Quality of English Language

N/A

Author Response

Title: Reference-Guided Flexible Gated Image Reconstruction Network for Multi-exposure Image Fusion

Journal title: Entropy

Manuscript ID: entropy-2770703

Authors: Yuhui Huang, Shangbo Zhou, Yufen Xu, Yijia Chen, Kai Cao

Dear Reviewer,

We extend our heartfelt gratitude for your meticulous review of our manuscript and the invaluable feedback you have offered. We deeply appreciate the guidance you provided in the initial two review rounds, and we have diligently incorporated relevant modifications in response to your insightful suggestions. During the third round of peer review, we have gone a step further to address your comments, effecting the following refinements:

Regarding the section in the ablation experiments where the improvement of the loss function achieves optimal performance, we provide the following explanation and annotate the modified parts in the latest manuscript:

We conclude that the improvement of the loss function (adding the gradient fidelity term) is necessary. We discuss whether the gradient fidelity term should exist and its balance parameter α in the loss function. We enhance Figure 12 (originally Figure 14) by adding examples of magnifying image details (text regions). It can be observed that when the gradient fidelity term is not introduced (α= 0), although the structural similarity index reaches its maximum level, the magnification of image details is not ideal, and the details of the text in the image appear blurry. As the balance factor α increases to 0.7, there is a significant improvement in the content and clarity of image details, but the metric based on structural similarity shows a certain degree of attenuation. However, this attenuation is meaningful, as the preservation of texture and high-frequency information in the image is crucial. When α reaches 0.9, the clarity and texture preservation of the image have reached a good level, and improvements in metrics based on image features, such as average gradient(AG) and edge intensity(EI), become limited. To avoid further compromising the image structure, we set the balance factor to 0.9. This decision considers the trade-off between the two aspects, aiming to preserve texture and details without sacrificing the main image structure.

In the latest manuscript, we provide a more explicit explanation of the reasons for the improvement of the loss function and the discussion of the balance parameter α. Additionally, we supplement examples of magnifying image details, demonstrating that this improvement is worthwhile even when sacrificing certain aspects of structural metrics like MEF-SSIMc or MEF-SSIM.

Overall, we believe that these revisions in the third round further enhance the quality of the manuscript, aligning it more closely with your expectations. We look forward to any additional guidance you may provide on our modifications and appreciate your invaluable suggestions for advancing the scholarly contribution. We welcome any further questions or suggestions you may have.

Thank you once again for your time and expertise.

Best regards,

Shangbo Zhou on behalf of the authors.

Corresponding author: Shangbo Zhou.

Email: shbzhou@cqu.edu.cn